# Localization of Major Ephedra Alkaloids in Whole Aerial Parts of Ephedrae Herba Using Direct Analysis in Real Time-Time of Flight-Mass Spectrometry

**DOI:** 10.3390/molecules26030580

**Published:** 2021-01-22

**Authors:** Nayoung Yun, Hye Jin Kim, Sang Cheol Park, Geonha Park, Min Kyoung Kim, Young Hae Choi, Young Pyo Jang

**Affiliations:** 1Department of Life and Nanopharmaceutical Sciences, College of Pharmacy, Kyung Hee University, Hoegi-dong 1, Dongdaemun-gu, Seoul 02447, Korea; nyyun@helixmith.com (N.Y.); sancho@khu.ac.kr (S.C.P.); ginapark0326@khu.ac.kr (G.P.); mindung3@khu.ac.kr (M.K.K.); 2Division of Pharmacognosy, College of Pharmacy, Kyung Hee University, Hoegi-dong 1, Dongdaemun-gu, Seoul 02447, Korea; hk2798@cumc.columbia.edu (H.J.K.); y.h.choi@biology.leidenuniv.nl (Y.H.C.); 3Natural Products Laboratory, Institute of Biology, Leiden University, Sylviusweg 72, 2333 BE Leiden, The Netherlands

**Keywords:** molecular imaging, direct analysis in real-time mass spectrometry, ephedra alkaloids, ephedrine, methylephedrine, *Ephedra sinica*

## Abstract

Mass spectrometry-based molecular imaging has been utilized to map the spatial distribution of target metabolites in various matrixes. Among the diverse mass spectrometry techniques, matrix-assisted laser desorption/ionization-mass spectrometry (MALDI-MS) is the most popular for molecular imaging due to its powerful spatial resolution. This unparalleled high resolution, however, can paradoxically act as a bottleneck when the bio-imaging of large areas, such as a whole plant, is required. To address this issue and provide a more versatile tool for large scale bio-imaging, direct analysis in real-time-time of flight-mass spectrometry (DART-TOF-MS), an ambient ionization MS, was applied to whole plant bio-imaging of a medicinal plant, Ephedrae Herba. The whole aerial part of the plant was cut into 10–20 cm long pieces, and each part was further cut longitudinally to compare the contents of major ephedra alkaloids between the outer surface and inner part of the stem. Using optimized DART-TOF-MS conditions, molecular imaging of major ephedra alkaloids of the whole aerial part of a single plant was successfully achieved. The concentration of alkaloids analyzed in this study was found to be higher on the inner section than the outer surface of stems. Moreover, side branches, which are used in traditional medicine, represented a far higher concentration of alkaloids than the main stem. In terms of the spatial metabolic distribution, the contents of alkaloids gradually decreased towards the end of branch tips. In this study, a fast and simple macro-scale MS imaging of the whole plant was successfully developed using DART-TOF-MS. This application on the localization of secondary metabolites in whole plants can provide an area of new research using ambient ionization mass spectroscopy and an unprecedented macro-scale view of the biosynthesis and distribution of active components in medicinal plants.

## 1. Introduction

Mass spectrometry-based molecular imaging has been used as a powerful analytical technique to visualize the distribution of endogenous and/or exogenous biomolecules in diverse tissues. This kind of image analysis can provide some information on the spatial location of metabolites, increasing the quality of the data obtained with conventional metabolomics. Using this technique, various molecules, such as lipids [1], proteins [2], peptides [2,3], and many specialized metabolites, e.g., flavonoids and glucosinolates [4], have been successfully imaged directly in the animal or plant tissue. It is Matrix-Assisted Laser Desorption/Ionization (MALDI) that, among many ionization techniques of mass spectroscopy, is undoubtedly one of the most popular ionization methods used in molecular imaging due to its efficiency in the ionization of large molecules, such as proteins and peptides. The popularity of MALDI for molecular imaging is due mainly to its uniquely high spatial resolution that makes micro-scale tissue imaging possible. Its use has been reported in many previous studies, such as the successful identification of candidates of tumoral markers of prostate cancer [5], the characterization of the spatial distribution of lens proteins and their modifications in lens sections [6], the knowledge of the distribution of neuropeptides in mouse pituitary glands [7], and understanding the distribution of specific drugs in tissue [8]. On the other hand, the limitations of MALDI imaging are well known. It requires a series of sample preparation steps, including embedding, section, mounting, matrix coating, and each step has a significant impact on the quality of the results [9].

Notwithstanding the potential of MALDI-based molecular imaging in mammalian tissues, there are only a few applications involving plants, and if any, these are limited to specific organs, such as leaves, stem, and grain tissue, which have been analyzed, for example, for their content of metabolites, such as flavonoid and saponins [10,11]. Thus, the use of MALDI techniques in plant imaging is mainly limited to a small area of the sample. For plant studies, particularly when molecular imaging is applied at a macro-scale to determine the distribution of metabolites at an organ level, the high spatial resolution power of MALDI limits the area of analysis due to the unacceptably lengthy sample preparation and analysis time needed for large-surface samples. Regarding sample preparation, MALDI analysis requires a step in which the tissue has to be covered with a matrix solution for the ionization and co-crystallized with this matrix [12]. The ionization efficiency of the molecules, however, varies according to the type of matrix [8].

As explained, the extension of the imaging area is an important criterion in plant research applications. This is especially crucial considering that mapping the distribution of compounds in a whole plant is necessary to obtain significant information about the biosynthesis, transportation, and biological functions of metabolites. With this in mind, different types of ionization sources were tested for the macro-scale molecular imaging to widen the analytical range.

A number of MS-methods based on ambient desorption ionization techniques have been developed to circumvent these limitations and increase the efficiency in terms of the time involved in the analysis, namely, direct analysis in real-time (DART), desorption electrospray ionization (DESI) [13], and atmospheric solids analysis probe (ASAP) [14]. Especially, DART has the edge over other techniques as the coverage of area and ease of sample preparation results in reduced analysis time. In the DART ion source, metastable helium gas reacts with atmospheric water to produce protonated water clusters, and the molecules on the sample surface are directly ionized by the transfer of protons from these water clusters [15]. Since the analysis takes place in open-air conditions, samples can be analyzed in diverse states or formats, i.e., as gases, liquids, solids, or powdered, without any specific sample pre-treatment. The ion source of DART can ionize various types of compounds directly from the surface of samples with little fragmentation, allowing molecular ion peaks to be efficiently detected from diverse surfaces, such as human skin, paper money, glass, thin layer chromatography (TLC) plates, clothing, and metals [16,17,18]. The DART ion source has, however, a relatively low spatial resolution of 1–2 mm as it is limited by the width of the helium gas beam [19]. Nevertheless, as an image screening tool, it is considered to have a high throughput capacity of analysis, which can be efficiently adapted to the large-scale analysis of samples, such as whole plants.

In this paper, we reported an application of the DART ion source to image contents of active components in a medicinal plant with the purpose of testing the feasibility of the DART ion source as a tool for the macro-scale localization of plants’ metabolites. The whole aerial part of a single *Ephedra sinica* (Ephedraceae), a plant traditionally used for weight loss and for its diaphoretic, stimulant, and antiasthmatic properties, was selected for this study. The major ephedra alkaloids—ephedrine/pseudoephedrine and methylephedrine/methylpseudoephedrine (Figure 1)—were imaged throughout the whole aerial part of this plant, and their spatial distributions were determined.

## 2. Results

In DART analysis, the temperature of helium gas largely affects the ionization sensitivity and data quality. Therefore, prior to the analysis of ephedra alkaloids in the samples, a number of experiments were implemented to determine its optimum temperature. The two targeted ephedra alkaloids were not clearly detected at 250 °C, and when the temperature was increased to 350 °C, some surface of the sample started to burn; thus, the temperature of helium gas was set to 300 °C for all the data measurement. The spatial resolution of the system employed in this study was evaluated by separate experiments using a standard solution on a TLC plate, and the maximum resolution was determined to be approximately 1.0 mm. The total experimental time of plant analysis was around 10 h that was still reasonable for the image analysis of the whole aerial part of a single ephedra plant.

Representative DART-TOF-MS spectra obtained from the outer and inner surface of both ephedra side branches and the main stem can be observed in Figure 2. The outer surface of the branch showed the strong peak of the protonated molecule [M + H]^+^ of methylephedrine and/or methylpseudoephedrine (*m*/*z* 180.1323) with a very weak peak of protonated ephedrine (Figure 2A). The inner section of the branch produced both peaks of protonated ephedrine/pseudoephedrine (*m*/*z* 166.1126) and protonated methylephedrine and methylpseudoephedrine (Figure 2B). The mass accuracies of observed mass values for the alkaloids are represented in Table 1. Interestingly, there was a large variation of the alkaloid concentrations according to the location since these alkaloids were barely detectable on both the outer and inner surfaces of the main stem, and the MS spectra showed a completely different pattern to those of the side branch (Figure 2C,D).

Molecular imaging of active components in a whole, single ephedra plant was successfully carried out by the integration of imaging data of the parts. Being diastereoisomers, ephedrine and pseudoephedrine and methylephedrine and methylpseudoephedrine have the same molecular mass. It was thus impossible to distinguish ephedrine and methylephedrine from their diastereomers only by MS analysis. Therefore, the total contents of the diastereomers pairs of ephedra alkaloids were used for molecular imaging. Imaging for total contents of ephedrine and pseudoephedrine (*m*/*z* 166) is presented in Figure 3. The green color (ion intensity of ca. 15 × 10^3^ and less) represented the lowest detectable intensity of alkaloids, and the color gradually changed to blue (ion intensity of 50 × 10^3^–90 × 10^3^), purple (ion intensity of 90 × 10^3^–110 × 10^3^), and red (ion intensity of ca. 120 x 10^3^ and more) as the ion intensity increased. Most parts of the outer surface of ephedra aerial parts appeared as green, while major parts of the inner surface were colored blue or red. These results mean that the content of ephedrine and pseudoephedrine in the inner section surface was much higher than on the outer surface. It was also observed that the side branches of the plant had a much higher content of ephedrine and pseudoephedrine than the main stem of the ephedra plant. Interestingly, it is not a woody stem but a side branch that is medicinally used in Korea [20]. The total content of ephedrine and pseudoephedrine decreased towards the end tip of the branch, where primary metabolism generating essential primary metabolites is believed to dominate over secondary metabolism, synthesizing ephedra alkaloids. As N-methyltransferase has been known as the key enzyme family in the biosynthesis of ephedra alkaloids, spatial existence and/or activities of these enzymes need to be evaluated to unveil the reason for the differential location of these alkaloids [21]. Molecular imaging for the total content of methylephedrine and methylpseudoephedrine (*m*/*z* 180) is presented in Figure 4. The molecular imaging of *m*/*z* 180 showed a similar pattern to that observed for ephedrine and pseudoephedrine. The content of methylephedrine and methylpseudoephedrine also showed higher intensity on the inner surface, decreasing towards the tip of the branch. In contrast to the images for ephedrine and pseudoephedrine, some ion peaks of methylephedrine and methylpseudoephedrine were observed on the outer surface, as represented by blue and red color. While the maximum ion intensity for ephedrine/pseudoephedrine was around 150,000, the maximum ion intensity for methylated ephedrine/pseudoephedrine was significantly higher (ca. 250,000), and this represented that the content of methylephedrine and methylpseudoephedrine was higher than the sum of ephedrine and pseudoephedrine. This result seems to be opposite to that reported in a previous study using HPLC in which the content of ephedrine and pseudoephedrine in Ephedrae Herba was reported to be about 10 times that of methylephedrine and methylpseudoephedrine [22]. Due to the characteristic of DART-MS, only the compounds present on the surface are ionized and detected, so the quantitative results cannot be compared to those from the whole extract of the sample. In this study, it is not possible to generalize this to the amount contained in the whole plant, as only the relative amount of alkaloids present in some particular areas can be obtained. Nevertheless, it is significant that the distribution of some alkaloids present on the outer and inner surfaces of the entire plant can be seen as a whole. It is a great advantage that the distribution of the compounds of interest in the whole plant can be mapped as an overall picture, even if the accuracy is compromised.

## 3. Discussion

In this work, we showed that rapid and simple macro-scale molecular imaging could be achieved using DART-TOF-MS. This could constitute an improvement in MALDI-MS bio-imaging, which requires a long time to scan large surfaces of a whole plant. On the contrary, this method could be used effectively to image the target compounds in whole plants in a relatively short time without the need of a specific chemical matrix nor a specific probe. This experiment broadens the area of applications of DART-MS, allowing the bio-imaging of the whole plant, thus providing an overall view of the distribution of metabolites in whole plants. The results of the implementation of this technique can also be used to gain a deeper understanding of biological or physiological functions of plant secondary metabolites, the identification of biological markers, biosynthesis, and the transportation of metabolites upon biotic and/or abiotic stress, among many other applications.

## 4. Materials and Methods

### 4.1. DART-TOF-MS Measurement of Ephedrae Herba

A single *Ephedra sinica* was harvested from the herbal garden of the College of Pharmacy, Kyung Hee University, and stored in a deep freezer (Thermo Fisher Scientific, Asheville, NC, USA) at −70 °C. A voucher specimen (KHUP-0801) was deposited in the Museum of Korean Traditional Herbal Medicines located in the College of Pharmacy, Kyung Hee University. The aerial part, branch, and woody stem were cut into 10–20 cm pieces, and each piece (diameter range: 2–5 mm) was cut in half vertically so that similar parts of the center were exposed. The measurement was carried out using a DART ion-source (IonSense, Sangus, MA, USA) coupled to an Accu-TOF-MS (JMS-T100TD, JEOL, Tokyo, Japan). The optimized MS operating conditions were as follows: samples were analyzed in the positive ion mode; voltage of first orifice lens was 15 V, ring lens voltage was 5 V, and helium gas flow rate was 3 L/min. Mass scale calibration was accomplished by introducing a glass capillary with polyethylene glycol 600 (PEG 600, Sigma-Aldrich, St. Louis, MO, USA) in the DART. The analyzer was set with a peak voltage of 1000 V, a bias voltage of 28 V, a pusher bias voltage of −0.55 V, and a detector voltage of 2200 V. Mass detection range was *m*/*z* 50 to 1000.

The helium gas (Shinyang, Seoul, Korea) temperature and sample module were optimized before molecular imaging. The temperature of the helium gas in the ionization source was chosen according to the ionization efficiency of the target alkaloids among six different temperatures, i.e., 100 °C, 150 °C, 200 °C, 250 °C, 300 °C, and 350 °C. Since there was no specific sample module for molecular imaging with the DART-MS, a house-made module was used for the analysis. A piece of dissected branch and stem of ephedra plant was fixed on a glass plate with double-sided adhesive tape and introduced into the ion source using the appropriate sample module. In order to acquire accurate and reliable MS data, the glass plate bearing the plant sample was introduced using a sample module on a rail under the helium gas flow. Among the three different sample modules available, the 12 DIP-it holder module (IonSense, Sangus, MA, USA) was able to move the plate consistently, exposing the surface of the plant sample just beneath the helium gas jet. Three slide glass plates had to be stacked and glued to align them exactly with the helium gas jet (Figure 5). The analysis was conducted on both the outer surface and inner surface of Ephedrae Herba to compare its contents in ephedra alkaloids. The rail speed was set to 0.2 mm/s.

### 4.2. Data Processing

The raw data resulting from the spectral analysis were processed for molecular imaging. For this, selected ion chromatograms for the targeted ephedra alkaloids were extracted. As protonated molecular ion ([M + H]^+^) is mainly generated in DART ionization, selected ion chromatograms of *m*/*z* 166 and *m*/*z* 180 were extracted for ephedrine/pseudoephedrine and methylephedrine/methylpseudoephedrine, respectively. The identities of alkaloids were confirmed by direct comparison of the experimental high-resolution mass number with their theoretical mass numbers. The extracted chromatograms were converted to an ASCII file table in which the intensity of specific ions was recorded at 0.4 s intervals. The average of ten data points, which corresponded to 0.8 mm of the sample, was used for the imaging and assigned different colors using a specific function in the Excel program known as ‘conditional formatting’, which assigns different colors according to their numerical value. Whole plant imaging was completed by applying the corresponding colors to the photo of the plant using Adobe Photoshop (Adobe CS6). Each herb part had a length of 10~20 cm, and 1250~2500 data points were acquired and processed for the imaging.

## Figures and Tables

**Figure 1 molecules-26-00580-f001:**
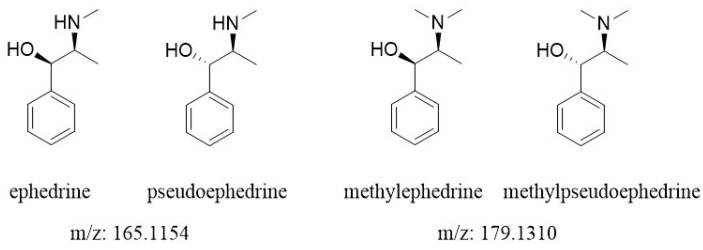
Major alkaloids of ephedra plants. Ephedrine and its diastereomer pseudoephedrine have the same molecular weight, so it is not differentiated by a mass spectrometer, and the same is true in the case of methylephedrine and methylpseudoephedrine. Ephedrine/pseudoephedrine: C_10_H_15_NO, methylephedrine/methylpseudoephedrine: C_11_H_17_NO.

**Figure 2 molecules-26-00580-f002:**
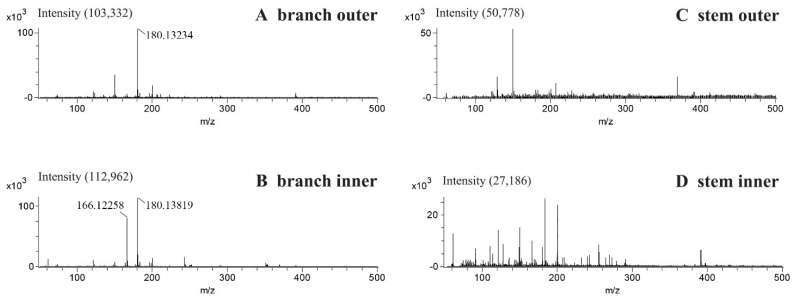
Representative direct analysis in real time-time of flight-mass spectrometry (DART-TOF-MS) spectra of the outer surface (**A**) and an inner surface (**B**) of a branch and outer surface (**C**) and an inner surface (**D**) of the main stem (woody stem) of ephedra aerial parts.

**Figure 3 molecules-26-00580-f003:**
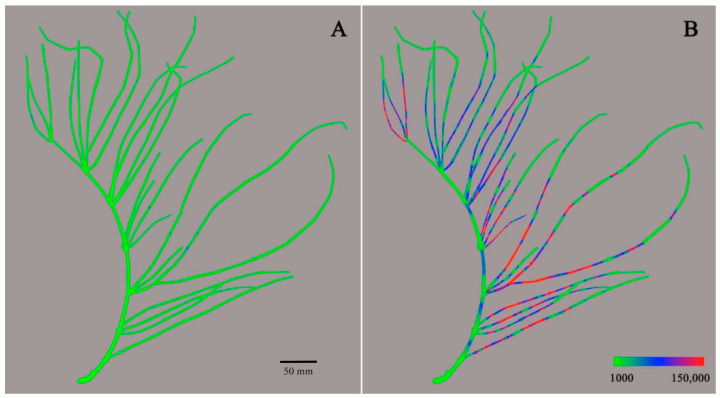
DART-MS images of total contents of ephedrine/pseudoephedrine (*m*/*z* 166) on the outer surface (**A**) and an inner surface (**B**) of ephedra aerial part. The color range is from 1000 to 150,000 ion intensity.

**Figure 4 molecules-26-00580-f004:**
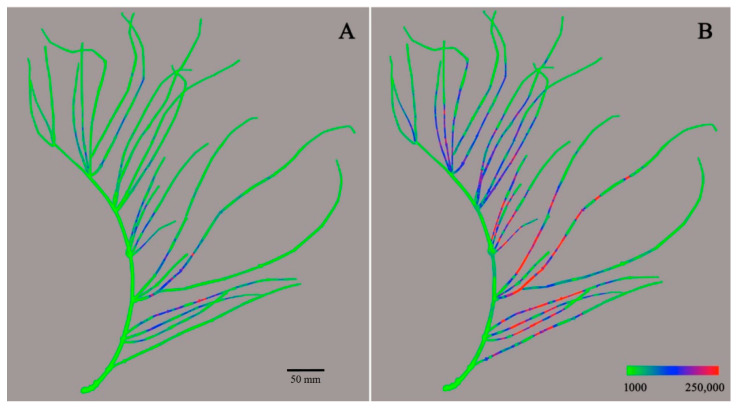
DART-MS molecular images for total contents of methylephedrine/methylpseudoephedrine (*m*/*z* 180) on the outer surface (**A**) and an inner surface (**B**) of ephedra aerial part. The color range is from 1000 to 250,000 ion intensity.

**Figure 5 molecules-26-00580-f005:**
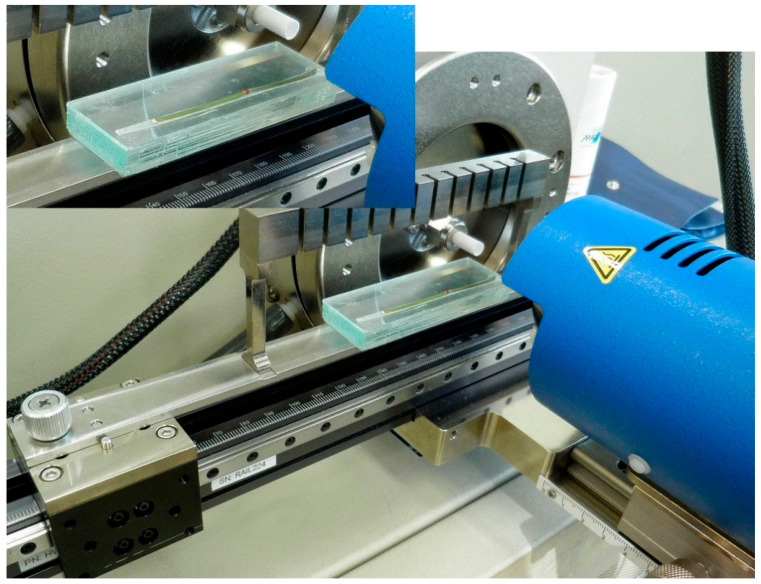
House-made sample module for DART-TOF-MS imaging system.

**Table 1 molecules-26-00580-t001:** Molecular formula, theoretical mass, and observed mass of major ephedra alkaloids and their protonated molecules. The mass difference was represented as millidalton (mDa).

Alkaloid	Molecular Formula	Theoretical Mass (Da)	Observed Mass (Da)	Mass Difference (mDa)
Ephedrine/Pseudoephedrine	C_10_H_15_NO	165.1154		
Protonated ephedrine	[C_10_H_15_NO + H]^+^	166.1227	166.1226	−0.1
Methylephedrine/Methylpseudoephedrine	C_11_H_17_NO	179.1310		
Protonated methylephedrine	[C_11_H_17_NO + H]^+^	180.1383	180.1323	−6.0

## Data Availability

The data presented in this study are available on request from the corresponding author.

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
