# Peer review of "Localization of Major Ephedra Alkaloids in Whole Aerial Parts of Ephedrae Herba Using Direct Analysis in Real Time-Time of Flight-Mass Spectrometry"

_molecules, 2021, doi:10.3390/molecules26030580_

Round 1

Reviewer 1 Report

This manuscript details the location of major alkaloids (Ephedrine associated compounds) within the whole ephedra herba plant using direct analysis in real time-time of flight-mass spectrometry (DART-MS). The results are very convincing and quite spectacular. Before publication, certain aspects of the manuscript however need to be addressed.

  • Throughout the manuscript, the English language is often approximative. Serious English language editing is required.
  • Line 59: I object to the words “…very complicated…”. Numerous teams worldwide (including mine) are successfully performing MALDI imaging MS experiments. Granted sample preparation has to be carefully controlled but it is not that complicated. Please rewrite.
  • Line 94: “…low spatial resolution…”, can you give numerical values?
  • Figure 2. Since only imaging MS of ephedrine associated compounds within a m/z range below 200 are presented here, I suggest restricting the m/z range of the figures to 0-500 (instead of 0-1000).
  • Line 142: “…and fragmentation pattern…”, this is not shown in the manuscript. Beyond accurate MW, could you fully characterize ephedrine and methylephedrin by MALDI MS/MS?
  • Lines 156-157: “…where primary metabolism is believed to dominate over secondary metabolism.” What are primary and secondary metabolisms? Please detail.
  • Line 181: Which “accuracy” are you referring to?
  • Figure 3, caption: When compared to the caption from Figure 4, the figure caption is incomplete. Please rewrite.
  • Figure 5: As it stands, the presented picture is minimally informative. Can you add a second picture to the figure zooming on the slide only?
  • Beyond whole plant images of ephedrine and methylephedrine, could other alkaloid/metabolite compounds be mapped? This would nicely complement this study. I suggest adding these results as supplemental data.

Author Response

  • Throughout the manuscript, the English language is often approximative. Serious English language editing is required.

: I carefully reviewed the entire manuscript according to the comments and corrected unclear expressions and grammatical errors. All modifications have been clearly marked in the revision.

  • Line 59: I object to the words “…very complicated…”. Numerous teams worldwide (including mine) are successfully performing MALDI imaging MS experiments. Granted sample preparation has to be carefully controlled but it is not that complicated. Please rewrite.

: "It requires a series of very complicated processes of sample preparation" was changed to "It requires a series of sample preparation steps" in the revised version.

  • Line 94: “…low spatial resolution…”, can you give numerical values?

: The numerical value was added to give more accurate information.

  • Figure 2. Since only imaging MS of ephedrine associated compounds within a m/z range below 200 are presented here, I suggest restricting the m/z range of the figures to 0-500 (instead of 0-1000).

: New figure 2 with m/z range of 0-500 was made and replaced.

  • Line 142: “…and fragmentation pattern…”, this is not shown in the manuscript. Beyond accurate MW, could you fully characterize ephedrine and methylephedrin by MALDI MS/MS?

: "and fragmentation pattern" was deleted in the revised version because we identified these alkaloids by their high resoultion mass. These mass numbers from Ephedra plant are generally accepted as from these alkaloids in phytochemistry.

  • Lines 156-157: “…where primary metabolism is believed to dominate over secondary metabolism.” What are primary and secondary metabolisms? Please detail.

"The total content of ephedrine and pseudoephedrine decreased towards the end tip of the branch where primary metabolism is believed to dominate over secondary metabolism" was changed to "The total content of ephedrine and pseudoephedrine decreased towards the end tip of the branch where primary metabolism generating essential primary metabolites is believed to dominate over secondary metabolism synthesizing ephedra alkaloids"

  • Line 181: Which “accuracy” are you referring to?

: We meant here that it is not appropriate to compare the quantitative HPLC result of the whole extract with our relative quantitative results from some parts of the plant surface. Nevertheless, we believe that our results are significant in that they show the relative distribution of major alkaloids across the whole Ephedra plant and this point is the main purpose of our study.

  • Figure 3, caption: When compared to the caption from Figure 4, the figure caption is incomplete. Please rewrite.

: The caption of Figure 3 was corrected.

  • Figure 5: As it stands, the presented picture is minimally informative. Can you add a second picture to the figure zooming on the slide only?

: The slide part was zoomed and add on the figure 5.

  • Beyond whole plant images of ephedrine and methylephedrine, could other alkaloid/metabolite compounds be mapped? This would nicely complement this study. I suggest adding these results as supplemental data.

: Bioimaging is possible for other ingredients in Ephedra plant. However, in this study, we reported this paper because ephedrine and its methyl derivatives, which are of the greatest interest both phytochemically and pharmacologically, being without overlapping molecular weight with other compounds, were considered the most suitable metabolites. We are sorry for not satisfying your suggestion, but please consider the practical issues that bioimaging on any other compounds with particular interest requires a considerable amount of work to produce another manuscript.

Reviewer 2 Report

The concerns raised by the reviewer have been addressed.

Author Response

Thanks for your kind review and decision.

Reviewer 3 Report

RE-REVIEW

Submitted to: Molecules

Manuscript Number: molecules-1070931 (previously molecules-1007778)

Title: Localization of Major Ephedra Alkaloids In Whole Aerial Parts of Ephedrae Herba Using Direct Analysis In Real Time-Time of Flight-Mass Spectrometry

Authors: Nayoung Yun et al.

I have previously reviewed this manuscript and am pleased with the changes that the authors have made.

There are two items that should be further addressed prior to publication:

1) There is a mistake with Theoretical mass in Table 1.

The theoretical masses of ephedrine/pseudoephedrine and methylphedrine/methylpseudoephedrine are correct (165.1154 and 179.1310 Da, respectively).

However, the masses of the protonated compounds are incorrect.

The formula for protonated ephedrine should be [C10H15NO+H]+ (and clearly shows the addition of a proton).  This is different than C10H16NO (as indicated in the manuscript) as the mass of a proton (1.007276 Da) is not the same as the mass of a hydrogen atom (1.007825 Da).  The theoretical mass of protonated ephedrine should be 166.1227 Da.

Similarly, the formula for protonated methylephedrine should be [C11H17NO+H]+ and the theoretical mass should be 180.1383.

Once the theoretical masses of the protonated compounds are correct, then the mass difference (mDa) will also need to be recalculated.

Also note – pseudoephedrine is also spelled incorrectly in Table 1 and units (Da) should be indicated for Theoretical mass and Observed mass.

2) The authors need to be clear to readers, throughout their manuscript, that this work involves the imaging of only a single plant. Omission of this detail is misleading. I would suggest the following changes:

Line 28 – “the whole aerial part of a single plant”

Line 98 – “The whole aerial part of a single Ephedra sinica…”

Line 117 – “and was reasonable for the image analysis of the whole aerial part of a single Ephdera plant”

Line 138 – “Molecular imaging of active components in a whole, single Ephedra plant was…”

Line 199 – “A single Ephedra sinica was harvested from the herbal garden…”

Author Response

1) There is a mistake with Theoretical mass in Table 1.

The theoretical masses of ephedrine/pseudoephedrine and methylphedrine/methylpseudoephedrine are correct (165.1154 and 179.1310 Da, respectively).

However, the masses of the protonated compounds are incorrect.

The formula for protonated ephedrine should be [C10H15NO+H](and clearly shows the addition of a proton).  This is different than C10H16NO (as indicated in the manuscript) as the mass of a proton (1.007276 Da) is not the same as the mass of a hydrogen atom (1.007825 Da).  The theoretical mass of protonated ephedrine should be 166.1227 Da.

Similarly, the formula for protonated methylephedrine should be [C11H17NO+H]and the theoretical mass should be 180.1383.

Once the theoretical masses of the protonated compounds are correct, then the mass difference (mDa) will also need to be recalculated.

Also note – pseudoephedrine is also spelled incorrectly in Table 1 and units (Da) should be indicated for Theoretical mass and Observed mass.

Thank you for your accurate comments. All points pointed out were reflected in the revised version.

2) The authors need to be clear to readers, throughout their manuscript, that this work involves the imaging of only a single plant. Omission of this detail is misleading. I would suggest the following changes:

Line 28 – “the whole aerial part of a single plant”

Line 98 – “The whole aerial part of a single Ephedra sinica…”

Line 117 – “and was reasonable for the image analysis of the whole aerial part of a single Ephdera plant”

Line 138 – “Molecular imaging of active components in a whole, single Ephedra plant was…”

Line 199 – “A single Ephedra sinica was harvested from the herbal garden…”

: Thanks for the considerate comments. All the comments were reflected in the revision.